# Clinical Presentations, Predictive Factors, and Outcomes of *Clostridioides* *difficile* Infection among COVID-19 Hospitalized Patients—A Single Center Experience from the COVID Hospital of the University Clinical Center of Vojvodina, Serbia

**DOI:** 10.3390/medicina58091262

**Published:** 2022-09-12

**Authors:** Nadica Kovačević, Dajana Lendak, Milica Popović, Aleksandra Plećaš Đuric, Maria Pete, Vedrana Petrić, Siniša Sević, Slavica Tomić, Jelica Alargić, Dimitrije Damjanov, Dijana Kosjer, Milica Lekin

**Affiliations:** 1Faculty of Medicine, University of Novi Sad, 21137 Novi Sad, Serbia; 2Clinic for Infectious Disease, University Clinical Center of Vojvodina, 21137 Novi Sad, Serbia; 3Clinic for Nephrology and Clinical Immunology, University Clinical Center of Vojvodina, 21137 Novi Sad, Serbia; 4Clinic for Anesthesiology, Intensive Care and Pain Therapy, University Clinical Center of Vojvodina, 21137 Novi Sad, Serbia; 5Emergency Department, University Clinical Center of Vojvodina, 21137 Novi Sad, Serbia; 6Clinic for Gastroenterology and Hepatology, Clinical Center of Vojvodina, 21137 Novi Sad, Serbia; 7Clinic for Endocrinology, Diabetes and Metabolic Disorders, University Clinical Center of Vojvodina, 21137 Novi Sad, Serbia

**Keywords:** *Clostridioides difficile* infection, COVID-19, antibiotics, risk factors

## Abstract

*Background*: This study aimed to investigate the clinical form, risk factors, and outcomes of patients with COVID-19 and *Clostridioides difficile* co-infections. *Methods*: This retrospective study (2 September 2021–1 April 2022) included all patients with *Clostridioides difficile* infection (CDI) and COVID-19 infection who were admitted to the Covid Hospital of the University Clinical Center of Vojvodina. *Results*: A total of 5124 COVID-19 patients were admitted to the Covid Hospital, and 326 of them (6.36%) developed hospital-onset CDI. Of those, 326 of the CDI patients (88.65%) were older than 65 years. The median time of CDI onset was 12.88 days. Previous hospitalizations showed 69.93% of CDI patients compared to 38.81% in the non-CDI group (*p* = 0.029). The concomitant antibiotics exposure was higher among the CDI group versus the non-CDI group (88.65% vs. 68.42%, *p* = 0.037). Albumin levels were ≤ 25 g/L among 39.57% of the CDI patients and 21.71% in the non-CDI patients (*p* = 0.021). The clinical manifestations of CDI ranged from mild diarrhea (26.9%) to severe diarrhea (63.49%) and a complicated form of colitis (9.81%). Regarding outcomes, 79.14% of the CDI patients recovered and 20.86% had fatal outcomes in-hospital. Although a minority of the patients were in the non-CDI group, the difference in mortality rate between the CDI and non-CDI group was not statistically significant (20.86% vs. 15.13%, *p* = 0.097). *Conclusions*: Elderly patients on concomitant antibiotic treatments with hypoalbuminemia and with previous healthcare exposures were the most affected by COVID-19 and CD co-infections.

## 1. Introduction

COVID-19 (a coronavirus disease) is a disease that has been among the greatest concerns of medical professionals worldwide for the last two years. After SARS-CoV-2 (severe acute respiratory syndrome-coronavirus) infection, most patients develop respiratory symptoms as the main clinical manifestation of the disease, but 3.8–34.0% of patients may also experience gastrointestinal symptoms, most often diarrhea. Previous research has shown that SARS-CoV-2 viral infection causes disruption of the intestinal microbiota and significant dysbiosis. On the other hand, SARS-CoV-2 virus-caused intestinal mucosa inflammation predisposed patients to infection with other intestinal pathogens such as *Clostridioides difficile* (CD) [1]. The CD infection (CDI) target risk population is elderly, immunocompromised, hospitalized patients with multiple comorbidities. In addition, the COVID-19 pandemic has seen an expansion of the overuse of antimicrobial drugs, which is the main cause of CDI in hospitals [2]. Although COVID-19 is a viral disease, research has shown that during the pandemic, over 70% of patients received broad-spectrum antibiotics to treat or prevent bacterial superinfections and coinfections [3,4]. CD is known to be a multidrug-resistant pathogen and a major cause of antibiotic-induced diarrhea. CDI is now considered one of the most significant nosocomial infections worldwide. The deterioration of the CDI epidemiological picture is mainly attributed to the emergence of a new virulent strain of CD, ribotype 027, and an increase in the aging population in many countries [5]. CDI symptoms range from mild diarrhea to very severe forms of fulminant colitis. The worrying fact is that the last decade has seen increased CDI incidence, a surge in severe disease forms, an increase in the frequency of relapses, and an increase in mortality rates. Studies have shown that despite currently available CDI therapy, as many as 15–35% of patients relapse after an initial disease episode [6].

The mechanisms causing diarrhea during COVID-19 viral infection are multiple and are largely conditioned by the virus itself, either directly or indirectly. The direct mode of gastrointestinal tract (GIT) cell infection occurs by the entry of the SARS-CoV-2 virus into the small intestine epithelial cells through ACE 2 receptors which are targeted by this virus and play a major role in intestinal inflammation regulation. In addition to directly damaging the intestinal barrier, a SARS-CoV-2 viral infection can affect the course of CDI by other mechanisms, such as intestinal microbial dysbiosis [7,8]. Studies have shown that the degree of intestinal dysbiosis is directly proportional to the level of the SARS-CoV-2 virus and to the severity of the COVID-19 viral infection [9]. The first data available from January 2020 to September 2021 indicated a reduced incidence of nosocomial CDI. Appropriate prevention measures introduced primarily due to COVID-19 are thought to have contributed to this significant reduction. Still, a significantly lower volume of testing for CDI has been observed, primarily due to diarrhea being viewed as part of the clinical picture of COVID-19 rather than of CDI. The inadequate interpretation of gastrointestinal symptoms during COVID-19 infection can lead to a late CDI diagnosis, delaying the introduction of adequate therapy and increasing the possibility of developing more severe forms of the disease [10,11]. Despite the facts from the various research mentioned above, many aspects of *Clostridioides difficile* and SARS-CoV-2 coinfection are still poorly understood.

The aim of our study was to evaluate the clinical characteristics, predictive factors, and outcomes of CDI among hospitalized patients with COVID-19.

## 2. Methods

We performed a retrospective, single center study from 2 September 2021 to 1 April 2022 that included adult patients with CDI and COVID-19 coinfection who were admitted to the Covid Hospital at the University Clinical Center of Vojvodina, Serbia. We collected demographic (sex and age) and epidemiological data, clinical characteristics, laboratory parameters (peripheral leucocyte count, serum creatinine levels, albumin levels, and serum C-reactive protein), previous hospitalizations, comorbidities (diabetes mellitus, chronic respiratory disease, chronic renal failure, liver disease, cardiovascular disease, malignancy, and neurological disease), medications given for COVID-19 infection, antimicrobial treatments before and during hospitalization, the length of a hospital stay, CDI onset and characteristics, treatment of CDI, and patient outcomes. The data was collected from the electronic medical records of patients admitted to the hospital.

The diagnosis of COVID-19 was confirmed by real-time polymerase chain reaction (PCR) from nasopharyngeal and oropharyngeal swabs. The case confirmation was obtained using the Rotary Nucleic Acid Extraction System (GeneRotex 96L) (Xi’an Tianlong Scienceand Technology Co., Ltd., Xi’an City, China) and the Gentier Real-Time Quantitative PCR (Gentier 96E) (Xi’an Tianlong Science and Technology Co. Ltd., Xi’an City, China).

The clinical form of COVID-19 infection was defined in accordance with World Health Organization (WHO) criteria [12] as follows: **mild pneumonia**—clinical signs of pneumonia (fever, cough, dyspnea, and fast breathing) but no signs of severe pneumonia, including an SpO_2_ of ≥ 90% on room air; and **severe pneumonia**—clinical signs of pneumonia (fever, cough, dyspnea, and fast breathing) plus one of the following: a respiratory rate of >30 breaths/min, severe respiratory distress, or an SpO_2_ of < 90% on room air, and increased inflammatory markers.

The diagnosis of CDI was based on the presence of diarrhea (≥ 3 watery stools within 24 h) associated with detection of the *C. difficile* toxin A or B. The etiology was confirmed by the enzyme-linked fluorescent essay (ELISA) and the RIDASCREEN *C. difficile* Toxin A and B (C0801), R-Biopharm AG, Germany. The testing was performed via glutamate dehydrogenase (GDH) and ELISA for the toxin and was considered diagnostic if both tests were positive [13]. All stool specimens from our study patients were cultured for the *Salmonella*, *Shigella*, *Yersinia enterocolitica,* and *Campylobacter species* to exclude other infectious causes of diarrhea.

CDI severity was defined in accordance with the European Guidelines for the treatment of CDI [14], as follows: **mild CDI**—absence of the following criteria: fever (>38.5°), hemodynamic instability, leukocytosis (leukocytes > 15,000 cells/μL), serum creatinine increase of > 1.5 times the values before infection, increase in serum lactates, histological evidence of pseudo-membranous colitis, and radiological evidence of ileus or ascites; **severe CDI**—the presence of at least one of the following criteria: fever (>38.5°), hemodynamic instability, leukocytosis (leukocytes > 15,000 cells/μL), serum creatinine increase of > 1.5 times the values before infection, increase in serum lactates, histological evidence of pseudo-membranous colitis, and radiological evidence of ileus or ascites; and **complicated CDI**—an episode of CDI complicated by toxic megacolon, intensive care unit hospitalization, sepsis, surgery, or death caused by CDI.

Hospital-onset CDI (HO-CDI) was considered if the symptom onset was > 72 h from hospital admission. COVID-19 patients with diarrhea and without microbiological evidence of CDI formed the control group.

The study was approved by the Ethics Committee of the Hospital (No. 99/2022). The study was conducted in accordance with the Declaration of Helsinki.

### Statistical Analysis

The descriptive statistical parameters are shown in standard statistical variables, arithmetic means ($\bar{X}$), standard deviations (SD), and interval values (maximum and minimum). Testing statistical significance was determined for parametric data using the t-test and for non-parametric using the *X*^2^ test, Fisher’s test, or Mann–Whitney’s test. Multivariate logistic regression analysis was used in the analysis of the influence of risk factors on CDI occurrence. Variables that statistically significantly affected the CDI occurrence in univariate analysis were included in a multivariable logistic-regression model with the determination of the odds ratio (OR) and a 95% confidence interval (CI). Statistical analysis was performed with the statistical package SPSS version 20.0. For all tests, a *p*-value of less than 0.05 was considered to be statistically significant.

## 3. Results

A total of 5124 COVID-19 patients were admitted to our COVID hospital during the study period from 2 September 2021 to 1 April 2022. We identified 326 of the 5124 (6.36%) COVID-19 patients that developed hospital-onset CDI.

The demographic and epidemiological data, clinical characteristics, comorbidities, laboratory findings (obtained within 48 h of hospital admission), and the outcomes of the 326 COVID-19 patients with CDI and 152 COVID-19 patients with non-CDI diarrhea are presented in Table 1.

Most of the patients were of the age category of over 65 years in both the CDI and non-CDI diarrhea groups (88.65% and 75.0%, respectively), with no significant difference between these two groups (*p* = 0.42). The mean age of the 326 patients with COVID-19 and CDI was 72 years, ranging between 43 and 95 years. Most of the patients in the CDI and non-CDI diarrhea groups were males (60.42% and 58.55%, respectively).

Regarding patient comorbidities, most of the CDI patients had a medical history of chronic underlying illness. The most common were chronic pulmonary disease in (125 of the 326 patients (38.34%)), malignant diseases (in 87 of the 326 (26.68%)), and diabetes (in 99 of the 326 patients (30.37%)). The influence of certain comorbidities did not show a statistically significant effect on the CDI occurrence, but in the CDI group, there were significantly fewer patients without comorbidities, with 27 of 326 compared to 28 of 152 non-CDI diarrhea patients without comorbidities (8.28% vs. 18.42, *p* = 0.037). Among the 326 CDI patients, 228 (69.93%) had data showing previous hospitalizations in the two months before the current admission, which was a statistically significant difference compared to previous hospitalizations in the non-CDI diarrhea patients (59 out of 152 patients (38.81%) (*p* = 0.029)).

With regard to medication administered for COVID-19 before admission, 286 of 326 (87.73%), 173 of 326 (53.06%), and 112 of 326 (34.35%) CDI patients received antibiotics, proton pump inhibitors (IPP), and steroids (dexamethasone or methylprednisolone), respectively.

Most of the CDI patients (286 out of 326 (87.73%)) were exposed to at least one (range 1–4) antibiotic prior to CDI diagnosis, and this was a significant difference between the CDI and non-CDI diarrhea patients (96 of 152 (63.16%) (*p* = 0.042)) (Table 1). Of those 286 CDI patients who were using antibiotics prior to hospitalization, 62.9% (*n* = 180) of them were treated with one antibiotic, 33.2% (*n* = 95) with two, 2.4% (*n* = 7) with three, and 1.4% (*n* = 4) of the patients were treated with four antibiotics prior to hospitalization (Figure 1), for a total of 407 antibiotics in 286 patients. The most common antibiotics prescribed for the outpatient treatment of COVID-19 were azithromycin (196–68.5%), third generation cephalosporin (103–36.0%), and levofloxacin (61–21.3%), while others (penicillin and second generation cephalosporins) were used in 47 patients (16.4%) (Figure 2). There was no difference in PP and steroid use before admission between the CDI and non-CDI diarrhea patients.

Regarding the medications administered during the hospital stays, 289 of the 326 (88.65%) CDI patients received broad-spectrum antimicrobials. The average duration of concomitant antimicrobial therapy was 13 days (range of 8–27 days). The number of patients on concomitant antibiotic use was significantly higher among the CDI patients (289 out of 326, 88.65%) compared to those without CDI (104 out of 152, 68.42%) (*p* = 0.037). The most common antimicrobial class was quinolones (*n* = 142, 49.1%), third generation cephalosporines (*n* = 58, 20.0%), carbapenems (*n* = 56, 19.4%), and glycopeptides (*n* = 42, 14.5%), while others (colistin and aminoglycosides) were prescribed less commonly (*n* = 22, 7.6%) (Figure 3). Carbapenems, glycopeptides, and colistin were prescribed to patients with confirmed concomitant bacterial infections, according to the isolates and antibiograms of the bronchoalveolar lavage, hemocultures, and urine culture tests. Proton pump inhibitor and steroid usage during hospitalization were common among both the CDI and non-CDI patients, but a statistical significance was not reached (80.37% vs. 72.36%, *p* = 0.764 and 66.56% vs. 69.07%, *p* = 0.828, respectively).

In our findings, there were no differences in the C—reactive protein (CRP) and creatinine levels between the CDI and non-CDI diarrhea patients on presentation. Univariate analysis showed that hypoalbuminemia (an albumin level of < 25 g/L) and a white blood cell count of ≥ 15 × 10^3^/μL were the laboratory parameters with a significant impact on CDI occurrence in COVID-19 patients (39.57% vs. 21.71%, *p* = 0.019 and 43.86% vs. 28.94%, *p* = 0.021, respectively).

Regarding COVID-19 severity, 68 of the 326 (20.86%) CDI patients had clinical signs of mild COVID-19 pneumonia and 258 of the 326 (79.14%) CDI patients presented signs of severe COVID-19, with no difference between the CDI and non-CDI groups (20.86% vs. 38.81%, *p* = 0.288 and 79.14% vs. 63.15%, *p* = 0.193, respectively).

Concerning CDI severity, among the 326 CDI patients, 87 (26.69%), 207 (63.49%), and 32 (9.81%) had mild, severe, and severe complicated CDI, respectively. CDI therapy was been administered in accordance with the European Guidelines for the treatment of CDI [14].

The most common treatment for CDI was oral vancomycin (250 out of 326 patients, 76.68%). The majority of these patients responded to the standard doses of oral vancomycin (125 mg every 6 h for 7 days) (212 out of 250, 84.8%). Of the 250 total patients treated with oral vancomycin, 38 (15.2%) patients needed a longer course of vancomycin therapy to achieve resolution. Oral metronidazole therapy was administered to 39 (11.96%) of the 326 patients, and 9 of these patients were transitioned to oral vancomycin because of treatment failure with metronidazole (9 out of 39, 23.07%). The remaining 32 of the 326 patients (9.81%) with complicated CDI received treatment with oral and rectal vancomycin (500 mg every 6 h), plus intravenous metronidazole (500 mg every 8 h). Most of these patients (28 out of 32, 87.5%) received therapy for 14 days to achieve resolution. No patients required colectomies for CDI. A total of 23 of the 326 CDI patients (9.74%) required intensive care unit admission in the setting of worsening status due to COVID-19.

The group of COVID-19 patients with CDI had a significantly longer hospital stay (25.68 ± 13.5 days) compared to the COVID-19 patients without CDI (14.34 ± 8.7 days) (*p* = 0.044).

Regarding outcomes, 258 out of 326 COVID-19 patients with CDI (79.14%) recovered and were discharged at home, and 68 out of 326 patients (20.86%) died during hospitalization. CDI was not the main cause of death in these patients. They had multiple comorbidities, and their deaths were related to the severe form of the COVID-19 disease. The overall the difference in mortality rate between the CDI and non-CDI diarrhea groups was not statistically significant (20.86% vs. 15.13%, *p* = 0.097).

### Risk Factors for the Onset of CDI in COVID-19 Patients

Table 2 shows the results of the logistic regression analysis to identify the factors associated with the likelihood of contracting CDI during COVID-19 infection. The multivariate analysis demonstrated that hospitalization in the preceding two months (OR: 2.364 (95% CI: 1.328–4.786), *p* = 0.021), the administration of antibiotics during the hospital stay (OR: 1.496 (95% CI: 1.039–1.961), *p* = 0.025), and an albumin level of ≤ 25 g/L (OR: 4.153 (95% CI: 2.368–6.412), *p* = 0.019) were independent risk factors for CDI development in patients with COVID-19.

## 4. Discussion

As one of the most common nosocomial infections, CDI has become a global health concern in the last decade. Diarrhea during COVID-19 occurs in some patients as part of the disease’s clinical picture. However, the overuse of broad-spectrum antibiotics observed during the COVID-19 pandemic may favor diarrhea of other etiologies such as CDI [10,11,15]. Previous research has indicated a significant increase in CDI incidence during the COVID-19 pandemic compared to the pre-pandemic period, from 2.6% to 10.9% [16]. The results of our study have shown that 6.36% of patients developed intrahospital CDI during COVID-19 treatment. Similar results have been published by Allegretti et al. (5.2%) and Gavrielatou et al. (4.5%) [3,17]. Cojocariu et al. reported a slightly higher CDI incidence (12.5%) [2]. The vulnerable population for CDI is elderly patients, as confirmed by our study, in which 88.65% of CDI patients were over 65 years of age.

Previous research has shown that repeated and prolonged hospitalizations are a significant risk factor for CDI in COVID-19 patients [2,18,19]. Lewandowski et al. have shown that the risk of contracting CDI increases by 3% with each day of hospitalization (OR = 1.03, 95% CI (1.01–1.05)) [16]. Accordingly, our study also showed that 69.93% of CDI patients reported hospitalization in the preceding two months before CDI, which was a statistically significant difference compared to the 38.81% of COVID-19 patients who did not contract CDI (*p* = 0.029). Similar results were published by Cojocariu, where 59.5% of CDI patients reported previous hospitalizations [2]. The conclusion that previous hospitalizations are a significant risk factor for CDI was confirmed by Marinescu et al. (*p* = 0.004) [20].

Even though it is a viral disease, the excessive and irrational use of antibiotics has certainly marked the COVID-19 pandemic [21,22]. Some studies show that 75% of CDI/COVID-19 patients took antibiotics before admission to the hospital [20]. In our study, the excessive use of antibiotics was also registered. Namely, our results show that 87.73% of COVID-19 patients were treated with antibiotics before developing CDI. Sehgal et al. point to a similar situation in their research [23,24]. With the inpatient treatment of COVID-19 infection, the antibiotics class in our study were quinolones (49.1%), third generation cephalosporines (20.0%), carbapenems (19.4%), and glycopeptides (14.5%). The last two of these were most often introduced for the concomitant treatment of bacterial pneumonia, sepsis, and urinary tract infections. The most commonly used antibiotics in the study by Granata et al. were beta-lactams, which were received by 52% of the patients with concomitant bacterial infections [19]. The use of other medications such as IPP and steroids was common in our patients. However, the difference in the use of these medications between the CDI and non-CDI diarrhea patients was not statistically significant. Still, some other authors emphasize that steroid therapy is a significant risk factor for the occurrence of CDI in the COVID-19 patient population [19].

Our study also analyzed laboratory parameters that may indicate an increased risk of developing CDI in COVID-19 patients. The results of the multivariate analysis showed that hypoalbuminemia is the only statistically significant risk factor for CDI (OR: 4.153 (95% CI: 2.368–6.412), *p* = 0.019). Similar conclusions were published by Granata [19]. These results are consistent with previous research showing that hypoalbuminemia is a sign of poor nutritional status, more severe concomitant chronic diseases, and poor host immune defense, which can significantly increase the risk of CDI [25].

The severity of SARS-CoV-2 and CDI coinfection is also indicated by immunological studies carried out during the pandemic, showing that cytokine production during CDI is very similar to cytokine production in patients with severe COVID-19. Namely, CD stimulates the production of a greater number of inflammatory cytokines that play a crucial role in both the pathogenesis of CDI and COVID-19, which leads to the possibility of developing more severe forms of coinfection [26,27,28]. As for the CDI clinical picture severity in our patients, 26.69% of patients had a milder form of CDI, 63.49% had a severe form, and 9.81% had a severe complicated form of CDI. Contrary to our results, in a study published by Granata et al., a significantly higher number of patients had a milder form of the disease (60.5%) [19]. Similar to our results, Marinescu et al. registered 20% of COVID-19 patients with a milder form of CDI, but in this study, as many as 45% of the patients developed a more severe, complicated form of CDI [20]. An increasing number of patients with severe forms of CDI in clinical studies indicates the importance of timely diagnosis and correct interpretation of gastrointestinal symptoms in COVID-19 patients. Late diagnosis and inadequate therapy can lead to severe and complicated forms of CDI [11].

Concomitant SARS-CoV2 and CD infection can result in a poorer disease outcome and increased mortality. In this regard, the mortality in our population of COVID-19 patients with CDI was 20.86%. This result is significantly lower than the previously published results by Sandhu et al., who reported a mortality rate of 44% in their patients [29]. However, most authors’ published results of mortality rates were similar to ours (28.9%, 22.5%, and 19%) [19,20,23]. CDI was not the leading cause of death in our patients with SARS-CoV2 and CD coinfection. These patients had multiple comorbidities and were of an older age. The poor outcomes were primarily due to COVID-19 infection exacerbation. Similar to our result, Granata also reported that 28.9% of CDI/COVID-19 patients died during hospitalization, but CDI was the leading cause of death in only one patient [19]. Marinescu also reported a 22.5% mortality rate, and CDI was the leading cause of death in two patients [20]. However, the fact that the total number of patients who died in these studies was small should be considered. Maslennikov et al. showed that the mortality rate in patients with CDI was higher than those with CD-negative diarrhea after the 29th day of illness [30].

Our results, however, did not show a statistically significant difference in disease outcomes in COVID-19/CDI patients compared to the group of COVID-19 patients with non-CDI diarrhea (mortality rate of 20.86% vs. 15.13%, *p* = 0.097). Although CDI was not the leading cause of death in our patients, we are of the opinion that it could certainly have contributed to the poor disease outcomes.

Finally, our study shows that the group of COVID-19 patients with CDI had significantly longer hospital stays (25.68 ± 13.5 days) compared to the COVID-19 patients with non-CDI diarrhea (14.34 ± 8.7 days) (*p* = 0.044), which, in addition to medical implications, has significant financial implications.

The results of our study contribute to the identification of COVID-19 patients who are at increased risk for developing a severe nosocomial infection such as CDI. Early identification allows for timely diagnosis and therapy, which ultimately contributes to a better disease outcome. The limitations of this study are primarily its retrospective design and the fact that the data were obtained from one hospital center. Due to the growing number of publications indicating the rise of the new epidemic strain CD 027, which produces more toxins, spreads faster, and causes more severe forms of CDI, our future research will focus on the PCR ribotyping of CD strains in our settings.

In conclusion, our findings support the statement that prolonged or repeated hospitalization, the administration of antibiotics during hospital stays, and hypoalbuminemia in laboratory findings indicate an increased risk of developing CDI in COVID-19 patients. Due to the long-term adverse effects of the SARS-CoV-2 virus on the gut microbiome and the chronic deterioration of patients within the Post-COVID syndrome, we can expect a higher number of patients at increased risk for CDI in the future. This is exacerbated by the growing trend of other nosocomial infections, which will inevitably lead to repeated and prolonged hospitalizations and the need for repeated antibiotic treatments. Such a situation entails the danger of the selection of CD strains that are highly resistant to currently available antimicrobial drugs. Therefore, attention should continue to be paid to prevention measures in hospitals and, primarily, the proper use of antibiotics in clinical practice.

## Figures and Tables

**Figure 1 medicina-58-01262-f001:**
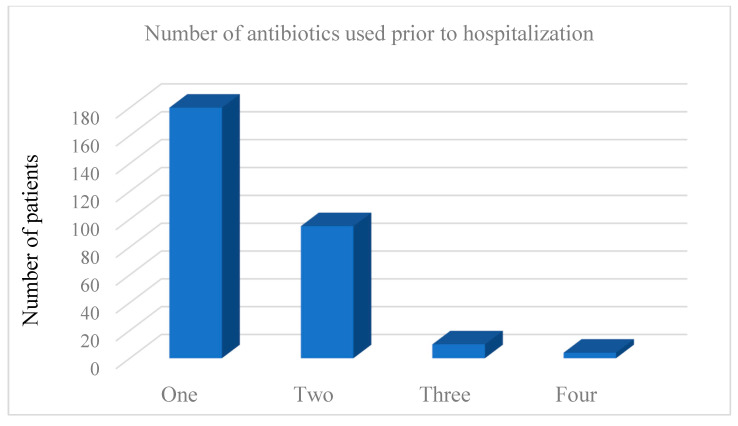
Number of antibiotics used prior to hospitalization.

**Figure 2 medicina-58-01262-f002:**
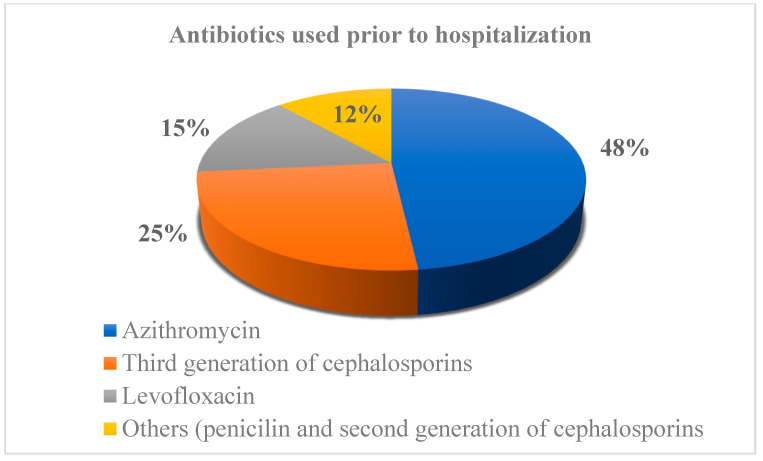
Antibiotics prescribed prior to hospitalization.

**Figure 3 medicina-58-01262-f003:**
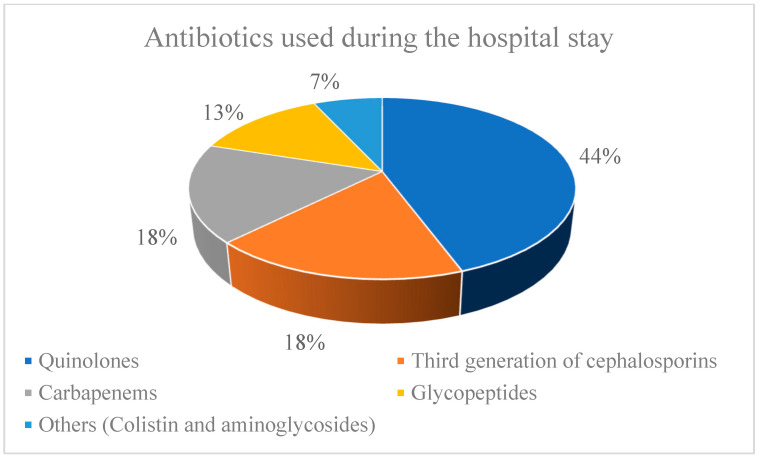
Antibiotics used during patient hospital stays.

**Table 1 medicina-58-01262-t001:** Demographic and epidemiological data, clinical characteristics, comorbidities, laboratory results, and outcomes of COVID-19 patients with and without CDI.

	CDI (*n* = 326)	Non-CDI Diarrhea (*n* = 152)	*p*-Value
**Gender (M)**	197 (60.42%)	89 (58.55%)	0.469
**Age > 65 years**	289 (88.65%)	114 (75.0%)	0.421
**Hospitalization in the previous two months**	228 (69.93%)	59 (38.81%)	0.029
**Diarrhea onset** (days)	12.88 ± 3.51	5.61 ± 2.95	0.028
**Comorbidities**			
No comorbidities	27 (8.28%)	28 (18.42%)	0.037
Cardiovascular disease	39 (11.96%)	25 (16.44%)	0.541
Diabetes	99 (30.37%)	50 (32.89%)	0.912
Chronic renal failure	50 (15.33%)	19 (12.50%)	0.769
Malignancies	87 (26.68%)	27 (17.76%)	0.063
Neurological disease	62 (19.01%)	26 (17.11%)	0.841
Chronic liver failure	43 (13.19%)	13 (8.55%)	0.443
Chronic pulmonary disease	125 (38.34%)	40 (26.31%)	0.296
Concomitant bacterial infections at admission	120 (36.81%)	39 (25.65%)	0.078
**Medication before hospital admission**			
Proton pump inhibitors	173 (53.06%)	84 (55.26%)	0.902
Antibiotics	286 (87.73%)	96 (63.16%)	0.042
Steroids ^+^	112 (34.35%)	57 (37.50%)	0.759
**Laboratory results**			
White blood cell count ≥ 15 × 10^3^/uL	143 (43.86%)	44 (28.94%)	0.021
Albumin ≤ 25 g/L	129 (39.57%)	33 (21.71%)	0.019
Creatinine ≥ 1.5 mg/dL	162 (49.69%)	71 (46.71%)	0.699
C-reactive protein mean (± SD)	129.44 (±48.69)	94.70 (±38.52)	0.076
**COVID-19 severity**			
Mild pneumonia	68 (20.86%)	59 (38.81%)	0.288
Severe pneumonia	258 (79.14%)	96 (63.15%)	0.193
**Medication during hospital stay**			
Proton pump inhibitors	262 (80.37%)	110 (72.36%)	0.764
Antibiotics	289 (88.65%)	104 (68.42%)	0.037
Steroids ^+^	217 (66.56%)	105 (69.07%)	0.828
**CDI severity**			
Mild	87 (26.69%)		
Severe	207 (63.49%)		
Severe complicated	32 (9.81%)		
**Total length of hospital stay** (days)	25.68 (±13.52)	14.34 (±8.74)	0.044
**Patient outcomes**			
Recovered	258 (79.14%)	129 (84.87%)	
Deceased	68 (20.86%)	23 (15.13%)	0.097

^+^ dexamethasone or methylprednisolone.

**Table 2 medicina-58-01262-t002:** Predictive factors associated with CDI occurrence during COVID-19 infection (multivariate analysis).

	OR Odds Ratio	95% CI Confidence Interval	*p*-Value
**Hospitalization in the preceding two months**	**2.364**	**1.328**–**4.786**	**0.021**
Antibiotics before hospital admission	1.203	1.042–1.969	0.193
**Antibiotics during hospital stay**	**1.496**	**1.039**–**1.961**	**0.025**
Diarrhea onset after COVID-19 diagnosis	0.759	0.284–1.128	0.367
White blood cell count of ≥ 15 × 10^3^/μL	0.187	0.053–0.654	0.084
**Albumin level of ≤ 25 g/L**	**4.153**	**2.368**–**6.412**	**0.019**

## Data Availability

The data presented in this study are available on request from the first author.

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
