# Peer review of "Clinical Presentations, Predictive Factors, and Outcomes of Clostridioides difficile Infection among COVID-19 Hospitalized Patients—A Single Center Experience from the COVID Hospital of the University Clinical Center of Vojvodina, Serbia"

_medicina, 2022, doi:10.3390/medicina58091262_

Round 1

Reviewer 1 Report

The authors have conducted a retrospective study evaluating the co-occurance of CDI and COVID-19 as a means to determine risk factors and outcomes in these patients. Based on their evaluation, the authors have met their goal by identifying risk factors like age, antibiotic use, and prior hospitalization that contributed to CDI post-hospitalization with COVID-19.

Clarification throughout the text with the term "non-CDI" would be helpful to readers. It is assumed thet "non-CDI" means the patients had diarrhea NOT caused by C. difficile. Please clearly define this term (perhaps at line 148 and with consistent language throughout the text) so that it is not misconstrued as describing patients who did not have diarrhea. The comparison between CDI-diarrhea and non-CDI-diarrhea is a very important one, and it is imperative that this is clear throughout the manuscript.

There are many grammatical and spelling errors that need corrected. A few instances are noted here, but this list is NOT allinclusive. Please have a native English speaker review for editing.

1. The term "second" written as a numeral should be written as "2nd" not "2th" (line 26). 

2. PPI has been reversed to IPP at line 182.

3. Subject/verb agreement is out of order in many cases throughout the manuscript.

4. Line 159: Regrding should be Regarding.

5. Line 190: Be complete and precise in language. Edit the following to be complete and without the "..." : "(colistine, amynoglicosides...)"

6. Define terms ONCE when first introduced. PPI has been defined twice. Once at line 171 and again at line 194.

There are two tables of data. It would be beneficial to report some of the key differences in risk factors in graph form. The reporting of antibiotic data (types of antibiotics, duration of antibiotics, number of antibiotics), for instance, could be more clearly represented in a bar graph complete with statistics. 

The presentation of risk factors in categories visualized in graph form would also be appealing.

Author Response

Thank you very much for careful reading and useful suggestions. 

Authors gave their best to improve the manuscript according to your suggestions.

"The authors have conducted a retrospective study evaluating the co-occurance of CDI and COVID-19 as a means to determine risk factors and outcomes in these patients. Based on their evaluation, the authors have met their goal by identifying risk factors like age, antibiotic use, and prior hospitalization that contributed to CDI post-hospitalization with COVID-19.

Clarification throughout the text with the term "non-CDI" would be helpful to readers. It is assumed thet "non-CDI" means the patients had diarrhea NOT caused by C. difficile. Please clearly define this term (perhaps at line 148 and with consistent language throughout the text) so that it is not misconstrued as describing patients who did not have diarrhea. The comparison between CDI-diarrhea and non-CDI-diarrhea is a very important one, and it is imperative that this is clear throughout the manuscript." Thank you for noticing, we have corrected it.

"There are many grammatical and spelling errors that need corrected. A few instances are noted here, but this list is NOT allinclusive. Please have a native English speaker review for editing." Certified translator corrected the language.

1. The term "second" written as a numeral should be written as "2nd" not "2th" (line 26). - corrected

2. PPI has been reversed to IPP at line 182. corrected

3. Subject/verb agreement is out of order in many cases throughout the manuscript. Certified translator corrected the language.

4. Line 159: Regrding should be Regarding. corrected

5. Line 190: Be complete and precise in language. Edit the following to be complete and without the "..." : "(colistine, amynoglicosides...)" corrected

6. Define terms ONCE when first introduced. PPI has been defined twice. Once at line 171 and again at line 194. corrected

There are two tables of data. It would be beneficial to report some of the key differences in risk factors in graph form. The reporting of antibiotic data (types of antibiotics, duration of antibiotics, number of antibiotics), for instance, could be more clearly represented in a bar graph complete with statistics. The presentation of risk factors in categories visualized in graph form would also be appealing. We have included 3 Graphs in our manuscript.

Thank you once again,

Authors

Reviewer 2 Report

Dear Authors,

You work on CDI co-infection in COVID-19 patients is very important, since the problem of CDI is constantly increasing and affecting wide range of patients. The work clearly states and proves with the data that previous hospitalization, exposure to the antibiotics are one of the key factors affecting co-infection by C. difficile.  The limitations of the study are well defined.

I think that the work would benefit very much if the ribotyping of the isolated C. difficile is included. I hope that this information is going to be included into follow up research.

In my opinion, the addition of the ideas how the problem of the co-infection can be avoided could have been included, although i understand that it is hard to make proposals in this case.

Kind regards

Author Response

Thank you very much for careful reading and suggestions for improvements. We are glad that you appreciate our efforts to collect and analyze data on this important subject. 

Unfortunately, we don't have data about ribotype, but for further investigations we are planing to include it. 

As you already mentioned, it is really hard to make proposals, except those that we have previously mentioned in the conclusion: reducing inapropriate antibiotic treatment would certainly reduce the incidence of CDI. 

Thank you in advance for supporting us,

Best regards,

Authors